# A Correction Method of Mixed Pesticide Content Prediction in Apple by Using Raman Spectra

**Yan Li [1] , Yankun Peng [1],*, Jianwei Qin [2] and Kuanglin Chao [2]**

[1] National Research and Development Center for Agro-processing Equipment, College of Engineering, China Agricultural University, Beijing 100083, China; liyan2017@cau.edu.cn

[2] USDA-ARS Environmental Microbial and Food Safety Laboratory, Building 303, BARC-East, Beltsville, MD 20705, USA; jianwei.qin@ars.usda.gov (J.Q.); kevin.chao@ars.usda.gov (K.C.)

\* Correspondence: ypeng@cau.edu.cn; Tel.: +86-10-6273-7703

**Abstract:** In the study, a new correction method was applied to reduce error during Raman spectral detection on mixed pesticide residue in apples. Combined with self-built pesticide residues detection system by Raman spectroscopy and the application of surface enhancement technology, rapid real-time qualitative and quantitative analysis of deltamethrin and acetamiprid residues in apples could be applied effectively. In quantitative analysis, compared with the intensity value of characteristic peaks of single pesticide with same concentration, the intensity value of characteristic peaks of the two pesticides decreased after mixing the pesticides, which affected the results severely. By comparing the difference in the intensity of characteristic peaks of single and mixed pesticides, a correction method was proposed to eliminate the influence of pesticides mixture. Characteristic peak intensity values of gradient concentration pesticide from 100 mg·kg$^{-1}$ to 10$^{-3}$ mg·kg$^{-1}$ and Lagrangian interpolation were applied in the correction method. And a smooth surface was applied to describe the correction coefficient of characteristic peak intensity. Through detecting the characteristic peak intensity values of the mixed pesticide, correction coefficient would be obtained. Then real values of the peak intensity of pesticides and the content of each component of the mixed pesticide would be acquired by the correction method. Correlation coefficient of model validation exceeded 0.88 generally and Root Mean Square Error also decreased obviously after correction, which proved the reliability of the method.

**Keywords:** Raman spectra; mixed pesticides; apple; correction method; rapid; real-time

## 1. Introduction

Pesticide plays an essential role in crop farming. Application of pesticide in agriculture has created a great deal of grain output and fed many people in the world. But this also caused some problems. Pesticides have chronic and acute impact to human health [1–3]. Pesticide poisoning to human health due to either processing during agricultural production or living poisoning that is because of consumption of food with high residuals, ingestion, residue and so on [4]. While in the detection of mixed pesticide residue, the interaction between different pesticide components always affects the intensity of characteristic peaks, which makes the detection results inaccurate. It's necessary to find a way to eliminate the interference. Deltamethrin ($C_{22}H_{19}Br_2NO_3$) is a kind of efficient synthetic pyrethroid insecticides which is created in 1974 with advantages like wide insecticidal spectrum, significant effect, low residue and so on. And deltamethrin has been widely applied in crops. While deltamethrin is also one of limited used pesticides. Some studies demonstrated that long-term intake of agricultural products with deltamethrin residue increased the risk of neurodegenerative disorders like Parkinson, Alzheimer disease, developmental deficits and learning disabilities [5–8]. Thus it's quite necessary to detect deltamethrin for reducing damage to the public safety. In continuous contact

to insecticides directly or indirectly may increases potential human health problems. It's necessary to reduce deltamethrin residue on fruits and vegetables to decrease contact. Relevant study on agricultural products had been done by some researchers. And on October 27th 2017, deltamethrin was classified as category 3 carcinogens by World Health Organization (WHO) International Agency for Research on Cancer [9]. Acetamiprid($C_{10}H_{11}ClN_4$) is a synthetic chlorinated nicotimine pesticide against insects that have gained resistance to ganophosphate, carbamate and synthetic pyrethroid. It has been widely used for agricultural pest control in many countries. In the experiment, long-term intake of high doses of acetamiprid will lead to breast cancer in adult mouse models and rib malformations in fetal mice. Acetamiprid also causes mutagenesis in human peripheral lymphocytes in vitro and synergistic mutagenesis effect with alpha-cypermethrin. In agricultural production, the alternate use or combination of deltamethrin and organophosphorus pesticides, such as non-pyrimid pesticides, is beneficial to slow down the development of pest resistance. Pesticide residue mixture of deltamethrin and acetamiprid residue always exists on crop. While compared with the characteristic peaks intensity of single deltamethrin and acetamiprid, the characteristic peaks intensity of the pesticides mixture is low in a certain range, which increases the difficulty in the detection.

Raman spectroscopy is defined as a spectroscopic and imaging technique that is used to record or observe vibration, rotational or other lower frequency modes in a system [10]. It possesses strong advantages compared to other analytical techniques [11]. It is based on Raman scattering effect or inelastic vibration and known for its ability to detect chemicals accurately and precisely [12]. As Raman fingerprint of substance is distinctly unique to each other, it owns the ability to discern different element in the material [13]. The water molecule contains only a very small single chemical bond, which makes its Raman signal very weak [14]. During the Raman detection no samples will be touched by chemical substance. It had already been used in quality control of pesticide formulations [15]. Tianfeng Xu developed a method to detect chlorpyrifos on apples through Raman spectroscopy technology [16]. And now Raman technology have been a fast, accurate method for non-destructive detection and in several cases in-situ method for detection of various materials is gaining its importance in agricultural application as well [17]. Surface enhanced Raman spectroscopy (SERS) technique, based on Raman scattering effect and combined with surface enhancement mechanism, is an analysis techniques with high sensitivity. It's one of the important developments of conventional Raman spectroscopy. It can not only detect the detailed structure information of the object, but also provide higher sensitivity with several orders of magnitude than the conventional Raman technology, which improves trace substances detection limit greatly.

In Raman spectra detection, it's obvious that in the case of pesticide samples with same concentration, the characteristic peak intensity of pesticide mixture was lower than the ones of single pesticide. The declining degree of peak intensity changes with the concentration of pesticides. The characteristic peak intensity reflected the concentration of pesticide. So the characteristic peak intensity of pesticide mixture could not reflect the content accurately. This increased the risk of misclassifying agricultural product with excessive pesticide as qualified product. Thus the phenomenon affected Raman spectral detection on pesticide mixture severely. In the study, acetamiprid and deltamethrin, two pesticides which were commonly used on apples, were the research objects. According to the Raman signal enhancement effect, silver sol was used as surface enhancer. And the exposure time, laser power, enhancement methods, and the number of collection points were optimized to get the best result. In order to find the declining degree and changing rule of peak intensity and pesticide concentration, the characteristic peaks intensity of single pesticide and pesticides mixture with gradient concentration were all detected and compared and the rates of intensity were calculated. Then the rates between single pesticide and mixture with concentration from 100 mg·kg$^{-1}$ to 10$^{-3}$ mg·kg$^{-1}$ were worked out with Lagrange interpolation method. A correction method was applied in Raman spectral detection of deltamethrin and acetamiprid residue. And a reasonable assessment of model validity was made. The correction method could be used in Raman spectral detection of other pesticides mixture.

## 2. Materials and Methods

### 2.1. Experimental Materials

In the experiment, pesticide solution with gradient concentrations of 100, 10, 1, $10^{-1}$, $10^{-2}$ and $10^{-3}$ mg·kg$^{-1}$ were made by deionized water and commercially available chlorpyrifos pesticides and acetamiprid pesticide with concentrations of 4800 mg·kg$^{-1}$ and 2000 mg·kg$^{-1}$ respectively. Before the detection, 50 mL of the solution was placed in a small mist spray gun. After washed with deionized water and dried naturally, apple samples were sprayed to simulate the spraying process of field pesticides to prepare experimental samples.

60 Fuji apples were bought as samples from the local market with different shape and color. 40 apples were chosen as calibration set and the others formed validation set. Experiment samples were prepared with the method of spray simulation as pesticide spraying process in the farmland. Before collecting Raman spectroscopy, 2 μL acetone was dropped on the surface of apple sample with diameter of 2 mm. Then 4 μL silver sol treated with centrifuge enrichment was dropped on the acetone drop rapidly with diameter of 3 mm. At last 1 μL nitric acid solution with concentration of 40 mmol·L$^{-1}$ was dropped on the mixture drop. Adding of acetone solution would extract pesticide in peel and disperse the pesticide molecule in the acetone solution, which made the silver nanoparticles and pesticide molecule fully mixed. Nitric acid shortened the distance between the pesticide molecules and the silver nanoparticles, which made the Raman signal enhanced. Each detection point was successively applied with acetone, silver sol and nitric acid solution. And then Raman spectra signal was collected. In order to get accurate Raman spectral information of apple samples, several detection points were detected from each apple sample. And the detection points were located over the entire surface of the sample.

After detecting the Raman spectra, the spectra of samples with same concentration were averaged as original spectrum at that concentration. Raman spectra of single pesticide and pesticide mixture with same concentration were detected from the same apple to reduce interference from apple background.

### 2.2. Pesticide Residue Detection System

Pesticide residue detection system developed by the lab was used to collect Raman spectral information in the study. The pesticide residue detection system included a laser device, Raman spectrometer, photoelectric charge coupled device, camera, optical fiber, probe, computer and other hardware. The effective detection range of the system was $-186.45{\sim}2325.72$ cm$^{-1}$. During the study, the excitation light power of the system was 500 mW with a wave length of 785 nm. The pesticide residue detection system was shown in Figure 1.

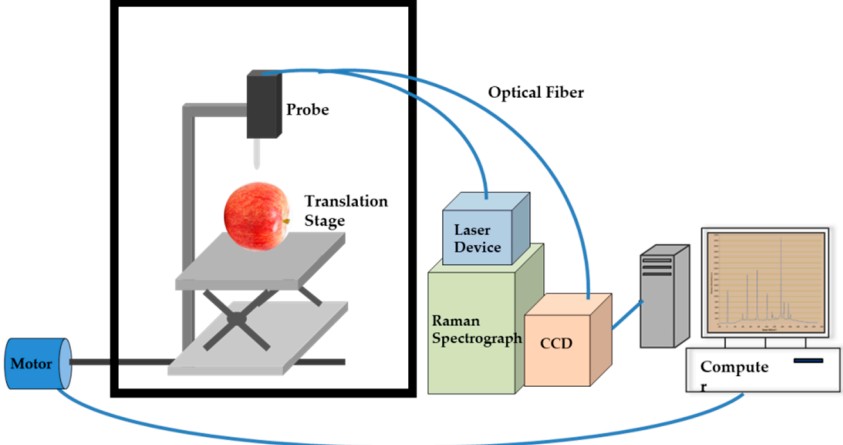

**Figure 1.** Pesticide residue detection system.

## 2.3. Preparation of Silver Sol

According to the method created by Nicolae Leopold et al. [18], a silver nitrate solution with a concentration of $1.1 \times 10^{-3}$ mol·L$^{-1}$ was prepared. And then hydroxylamine hydrochloride and sodium hydroxide were dissolved in 10 mL of deionized water to prepare a mixed solution with concentration of $1.5 \times 10^{-2}$ mol·L$^{-1}$ and $3.0 \times 10^{-2}$ mol·L$^{-1}$ respectively. The freshly prepared mixed solution was rapidly poured into 90 mL of rapidly stirred silver nitrate solution, and the color of the solution rapidly changed from transparent to gray-brown. And the stirring was continued for 30 min. Then the silver sol was obtained. The obtained silver sol was centrifuged. Then the supernatant was removed. At last the concentrated silver sol was preserved in dark environment at 4 °C for use.

## 2.4. Experiment Method

During the detection, apple sample was placed on a dedicated holding device. The surface of the apple was a curved surface, and the distance from vertex of apple surface to the Raman probe was kept at 7.5 mm. 9 detection points were taken in the equator line of each apple sample. 9 detection points were taken at random and each point was detected 3 times for representative signal. Among them, the relevant parameters of the detection system were: the laser wavelength is 785 nm, and the laser power was 450 mW. Integration time of CCD camera was an important parameter in Raman spectrum detection. The camera integration time was 3 s in the study.

## 3. Results and Discussion

### 3.1. Optimization of Exposure Time and Laser Power

Appropriate parameter values were essential in the detection. The main parameters included exposure time and laser power. For samples with high pesticide concentration, the signal of characteristic peaks was easy to obtain. However, when the concentration of samples was low, the signal of the characteristic peaks would be weak or disappear if the exposure time was short or the laser power was small. While too long exposure time and too great laser power would lead to signal saturation or samples being burned by laser. Longer exposure time would also lead to time wasting for sample detection and may cause damage to charge-coupled device (CCD) camera. Thus it was necessary to optimize exposure time and laser power before the detection. Different exposure time and power were applied for detecting samples with same concentration. It can be seen that when the laser power was certain, the longer the exposure time was, the stronger the characteristic peak signal was. And when the exposure time or the laser power was short, the signal of the characteristic peaks could hardly be observed. Finally when the exposure time was 3s and the laser power was 450 mW, the signal of characteristic peak was the strongest. The maximum power of the laser was 450 mW. Laser of 450 mW would not damage the apple samples. Hence the laser was set at 450 mW. If the exposure time exceeds 3 s, the signal may saturate and the CCD camera may be damaged.

### 3.2. Data Preprocessing and Analysis

When Raman spectroscopy was used for sample analyze, signal noise and fluorescence background were important factors which affect the accuracy of the analysis severely. The noise of the instrument, external environment, etc. leads to signal noise. In this study, the Savitzky-Golay (S-G) 5-point smoothing method [19] was used to remove the spectral noise. This method was proposed by Savitzky and Golay and was widely used in data stream smoothing and noise reduction. And the method was a filtering method based on local polynomial least squares fitting in the time domain. The biggest advantage of the method was that it could ensure the shape and width of the signal while filtering out the noise.

Fluorescence background interference was the most important factor in Raman signal analysis especially for organic or biological samples, which made signal of the target analyte submerged. And accurate and effective removing fluorescence background was very important in Raman analysis. The principle of adaptive iterative reweighted penalty least squares (airPLS) was to control the fidelity

and roughness of the fitting curve by weighting coefficients so as to obtain an ideal fitting curve in this study [20,21]. This method was time-saving and flexible. The weight of the overall variance between the baseline of the fit and the original signal was changed through iteration. The overall variance weight was acquired from the difference between the baseline and the original signal. The first derivative and second derivative methods could eliminate the interference from the baseline and other backgrounds effectively and improve the resolution and sensitivity. But the method may lead to an increase in the signal-noise ratio. The standard normal variable transformation could eliminate the Raman spectrum noise. It could not remove the fluorescence background interference due to the power change of the laser light source and the attenuation of the light intensity. Baseline calibration deducted the fluorescence background effectively and preserved the original spectral information and eliminated the effects of the instruments. Two baseline calibration methods, the polynomial fitting method of 8 times and the signal minimum maxima, were used in the study. Adaptive scaling was a common calibration method.

### 3.3. Pesticide Mixture Signal Analysis

Surface-Enhanced Raman Scattering (SERS) signal acquisition was applied to standard solution of acetamiprid and deltamethrin pesticide and mixture of the two pesticides. There were literatures reported that main characteristic peaks of acetamiprid pesticide were located at 634, 1114 and 2164 $cm^{-1}$, and the characteristic peaks of deltamethrin were located at 574, 735 and 1380 $cm^{-1}$. Spectrum of characteristic peaks of pesticide mixture change with gradient concentration was shown in Figure 2. In the spectrogram, the characteristic peaks of the two pesticides were obvious and the intensity of the characteristic peaks changed with the concentration of pesticides consistently. Thus it was certain that peaks at 634, 1114, 2164 $cm^{-1}$ and 574, 735, 1380 $cm^{-1}$ were characteristic peaks of acetamiprid and deltamethrin pesticide. Overlapping peak didn't exist between acetamiprid and deltamethrin.

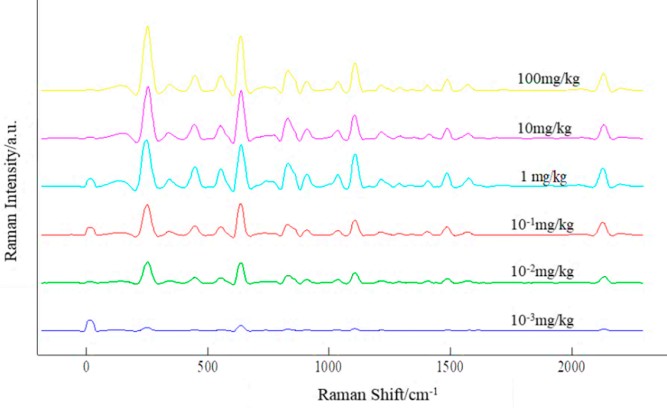

**Figure 2.** Spectrum of pesticide mixture characteristic peaks change with gradient concentration.

In Raman spectral detection, the characteristic peak intensity reflects the concentration of samples. Thus quantitative analysis could be made from Raman spectral signal. With the decrease of pesticide content, the intensity values of deltamethrin and acetamiprid characteristic peaks reduce linearly. As shown in Figure 2.

The location of the Raman vibration peak was only related to the vibration frequency of the chemical bond. Raman spectral peaks in different locations represented different chemical bonds [22]. According to common Raman spectral characteristic peak attribution map and comparison with other results analyze, it was possible to acquire the belonging of some characteristic peaks. Characteristic peaks of acetamiprid at 634, 1114 and 2167 $cm^{-1}$ belonged to C-Cl retraction, ring vibration, and ring "breathing" respectively. Characteristic peaks of deltamethrin at 574, 735, 1380 $cm^{-1}$ were caused by C-Br retractable, symmetrical $CBr_2$ flexing and ring scaling. The analysis results were consistent with the study by Dong [23]. And apple samples would cause no obvious Raman spectral peaks [24].

As deltamethrin and acetamiprid were always be used in the same agricultural season, it was necessary to detect the mixture of the two pesticides. It could be seen clearly from Figure 3 that intensity of characteristic peaks of deltamethrin and acetamiprid pesticide mixture at 634, 1114 and 2167 cm$^{-1}$ was lower than the ones of single deltamethrin or acetamiprid pesticide solution, which affected the accuracy of detection result severely. It was clear that the intensity of characteristic peaks decreased when the other pesticide was added. Table 1 showed the detection results comparison of gradient concentration mixed pesticides concentration and real pesticides concentration. Through analyzing original spectra of the two pesticides, it could be found that decrement of the peak intensity was in a certain extent. And the extent depended on the concentration of the two pesticides, exposure time of laser and integration time of CCD camera, mainly the concentration of pesticides. Therefore it was necessary to find out the link between the decrement extent of characteristic peaks and concentration of pesticides, and then to get the actual detection result.

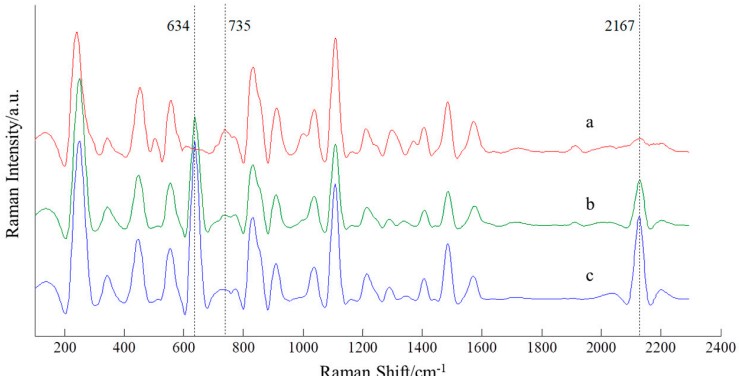

**Figure 3.** Spectrum of pesticide mixture and single pesticide with concentration of 100 mg·kg$^{-1}$. a: spectrum of deltamethrin sample; b: spectrum of deltamethrin and acetamiprid mixture. c: spectrum of acetamiprid sample.

**Table 1.** Detection results comparison of mixed pesticides concentration and real concentration.

| Pesticide Concentration (mg·kg$^{-1}$) | 100 | 10 | 1 | $10^{-1}$ | $10^{-2}$ | $10^{-3}$ |
|---|---|---|---|---|---|---|
| **T Value** | 0.016 | 0.031 | 0.039 | 0.046 | 0.087 | 0.365 |
| **RSD (%)** | 17.31 | 13.26 | 10.66 | 8.87 | 6.35 | 3.79 |
| **Standard Deviation** | 17.33 | 1.69 | 0.145 | 0.0113 | $7.67 \times 10^{-4}$ | $2.82 \times 10^{-5}$ |
| **Average Value (mg·kg$^{-1}$)** | 83.29 | 8.37 | 0.868 | 0.0899 | $9.38 \times 10^{-3}$ | $9.79 \times 10^{-4}$ |

By comparing a number of Raman spectra, it was clear to find that both characteristic peaks intensity value of single pesticide and pesticide mixture were steady in a range, which meant the multiple difference between single pesticide and pesticides mixture was certain. It was possible to find the ratio between characteristic peaks intensity value of single pesticide and the ones of pesticide mixture. Therefore the real peak intensity of one pesticide could be acquired through peak intensity of pesticides mixture and the ratios. The method of ratio would be realiable in Raman spectral detection of deltamethrin and acetamiprid mixture.

60 Fuji apples were used for detecting Raman spectral signals. 600 sample signals of deltamethrin, acetamiprid and the mixture of them were acquired. The Raman spectral information of deltamethrin, acetamiprid and the mixture of them with same pesticide concentration were detected from same apple sample to reduce the effect of sample background. Through analyzing massive full-spectrum Raman signal of deltamethrin and acetamiprid it was found that signal of characteristic peaks at 574, 1380 and 2167 cm$^{-1}$ was not strong enough for quantitative analysis. Thus characteristic peaks at 634, 735 and 1114 cm$^{-1}$ were used for data analysis of pesticide mixture.

### 3.4. Establishment of Regression Models

The mixture pesticides be detected included the deltamethrin and acetamiprid mixtures with same content and the ones with different concentration. As most pesticide mixture applied in farmland was in low content, detailed analyze of correction coefficient change rule was focused in low concentration. Deltamethrin, acetamiprid with concentration of 100, 10, 1, $10^{-1}$, $10^{-2}$ and $10^{-3}$ mg·kg$^{-1}$ were mixed as mixture in pairs. And there were 36 mixed mode. At least 15 sample points would be detected to get the average as representative concentration in each mixed mode to make sure the representative.

The characteristic peak intensity of the pesticide detected directly cannot be directly used to predict the concentration of the mixed pesticide. By analyzing the reduction ratio of mixed pesticides characteristic peak intensity compared with the intensity of single pesticides, a model of mixed pesticides peak intensity correction coefficient was established. Firstly, the concentration of the mixed pesticide was estimated by a single pesticide quantitative analysis model to get the estimated pesticide concentration. And then the estimated pesticide concentration was applied in correction coefficient model to acquire the correction coefficient. The characteristic peak intensity of the pesticide was multiply by the correction coefficient to get the corrected peak intensity. Finally the corrected pesticide concentration was obtained by applying corrected peak intensity to quantitative analysis model of the pesticide.

It was clear that correction coefficient change rule was the key to get accurate pesticide concentration. The correction coefficient was obtained by dividing the characteristic peak intensity value of a single pesticide by the characteristic peak intensity value of the pesticide mixture with same concentration. As quadratic function relations existed between pesticide concentration and the correction coefficients, two quadratic equations were needed to describe the change rule between content of pesticide mixture and the correction coefficients. And the change rule of one characteristic peak intensity was described by a binary quadratic polynomial.

As shown in Figure 4, the fitting effect of mixed pesticides concentrations and the correction coefficients was a smooth and slanted surface. For the characteristic peaks of acetamiprid at 634 cm$^{-1}$ and 1114 cm$^{-1}$, the correction coefficient reduced with not only the increments of acetamiprid concentration, but also the reduction of deltamethrin concentration. And when the concentration of one pesticide was low, its correction coefficient would be high, which indicated that components with low concentration in mixed pesticides were more likely to be affected by pesticides high concentration. For the change rule fitting model of peak in 634 cm$^{-1}$, the sum squared residual (SSE) was 0.0678. Root Mean Square Error (RMSE) and coefficient of correlation were 0.0475 and 0.9328 respectively. And for the change rule fitting model of peak in 1114 cm$^{-1}$, the SSE was 0.0328. RMSE and coefficient of correlation were 0.0330 and 0.8883 respectively.

A polynomial regression equation (Equation (1)) was acquired from the change rule fitting surface of characteristic peaks at 634cm$^{-1}$. In Equation (1), the concentrations of acetamiprid and deltamethrin were all from 100 mg·kg$^{-1}$ to $10^{-3}$ mg·kg$^{-1}$.

$$Z = 1.264 - 0.02345A + 0.01906D + 0.0006864A^2 - 0.0001839AD + 1.289 \times 10^{-5}D^2 \qquad (1)$$

where Z is the correction coefficient, A is the estimated concentration value of acetamiprid, D is the concentration value of deltamethrin.

The polynomial regression equation (Equation (2)) of characteristic peaks at 1114 cm$^{-1}$ was also obtained and shown below. In Equation (2), the concentrations of acetamiprid and deltamethrin were all from 100 mg·kg$^{-1}$ to $10^{-3}$ mg·kg$^{-1}$.

$$Z = 1.235 - 0.02271A + 0.01886D + 0.0007016A^2 - 0.0001841AD + 1.296 \times 10^{-5}D^2 \qquad (2)$$

where Z is the correction coefficient, A is the estimated concentration value of acetamiprid, D is the concentration value of deltamethrin.

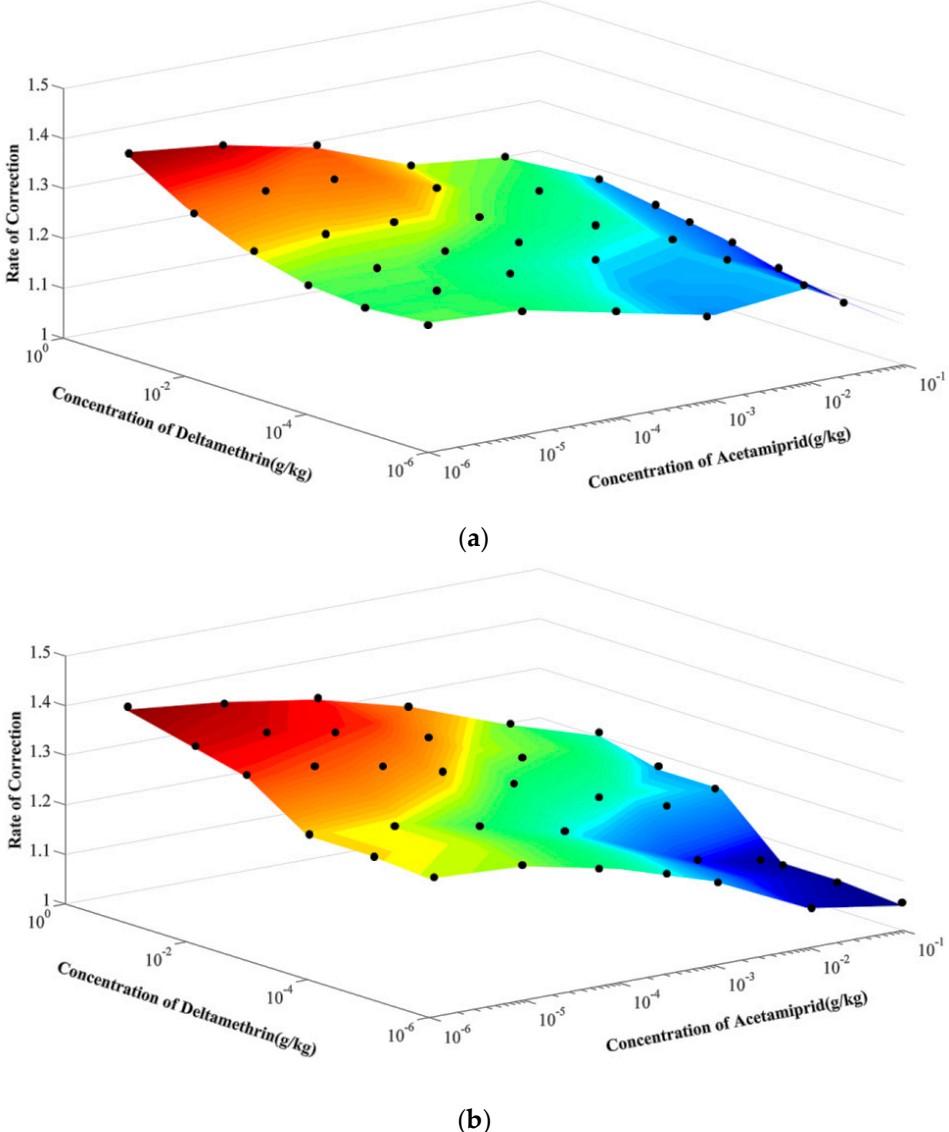

**Figure 4.** The fitting surface of characteristic peaks change rule at 634 cm$^{-1}$ (**a**) and 1114 cm$^{-1}$ (**b**); between pesticide concentration and correction coefficient.

The fitting surface of characteristic peak at 735 cm$^{-1}$ was shown in Figure 5. And for the characteristic peaks of deltamethrin at 735 cm$^{-1}$, the correction coefficient increased with the increase of acetamiprid concentration and the decrease of deltamethrin concentration. With the increase of acetamiprid content and decrease of deltamethrin content, the correction coefficient could be up to 1.32. The sum squared residual (SSE) of the fitting model was 0.0438. Root Mean Square Error (RMSE) and coefficient of correlation were 0.0382 and 0.9266 respectively.

The polynomial regression Equation (Equation (3)) of characteristic peaks at 735 cm$^{-1}$ was acquired and shown below. In Equation (3), the concentrations of acetamiprid and deltamethrin were all from 100 mg·kg$^{-1}$ to 10$^{-3}$ mg·kg$^{-1}$.

$$Z = 1.167 + 0.02066A - 0.002236D - 0.0006892A^2 - 8.851 \times 10^{-6}AD - 2.88 \times 10^{-6}D^2 \tag{3}$$

where Z is the correction coefficient, A is the concentration value of acetamiprid, D is the concentration value of deltamethrin.

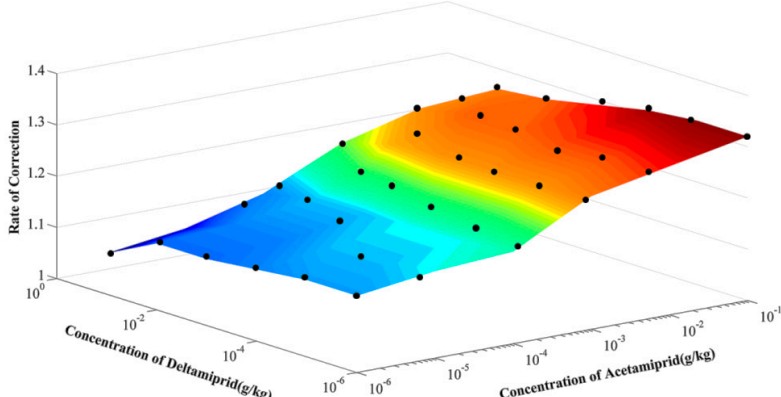

**Figure 5.** The fitting surface of characteristic peak change rule at 735 cm$^{-1}$ between pesticide concentration and correction coefficient.

### 3.5. Validation of the Correction Models

An experiment had been done to verify the reliability of the regression model. In the verification test of regression model, 200 deltamethrin and acetamiprid residue signals from 20 Fuji apples was set as validation set. The intensity of the characteristic peaks at 634, 735 and 1114 cm$^{-1}$ was extracted from the Raman spectrum with pretreatment. Then the intensity values would be applied to the pesticide residue quantitative model to get the estimated pesticide concentration. Correction coefficient would be acquired after substituting peak intensity into the regression model. And corrected peak intensity would be obtained through multiplying the peak intensity values and correction coefficient. After substituting corrected peak intensity into pesticide residue quantitative model, corrected pesticide concentration would be got. Through comparing the pesticide concentration and corrected pesticide concentration, a series of parameters would be obtained to judge the regression model.

After above steps, Raman signals of validation were processed. The modeling effect was evaluated by T value, correlation coefficient (R$^2$) and Root Mean Squared Error (RMSE). The results were shown in Figure 6. In Figure 6, "●" and "■" stood for the pesticide data before and after correction respectively.

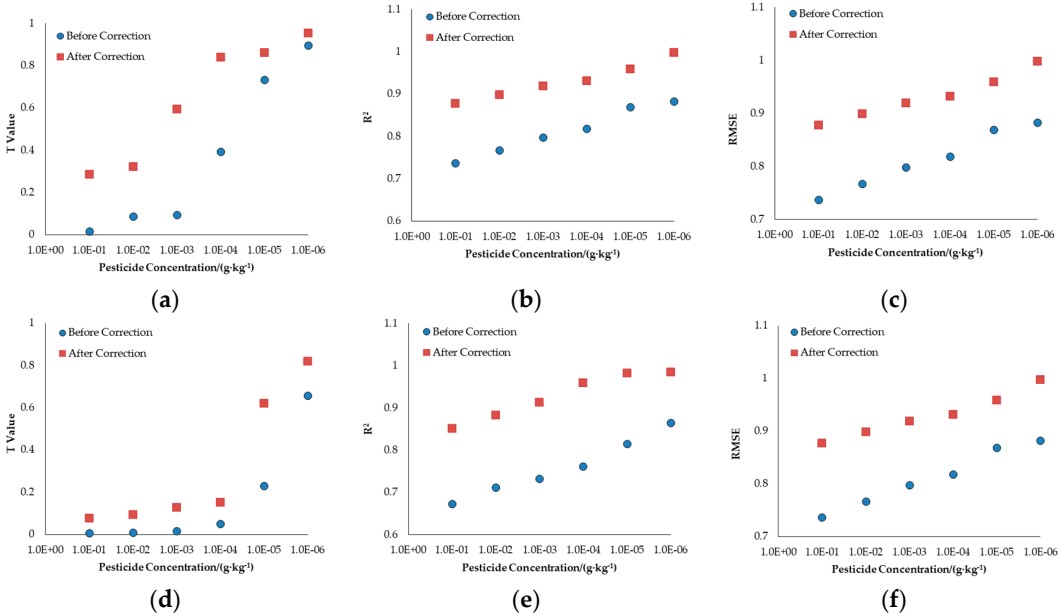

**Figure 6.** Results of regression model between pesticide concentration and T value, R$^2$ and Root Mean Square Error (RMSE). (**a**): T value of acetamiprid; (**b**): R$^2$ of acetamiprid; (**c**): RMSE of acetamiprid; (**d**): T value of deltamethrin; (**e**): R$^2$ of deltamethrin; (**f**): RMSE of deltamethrin.

From the figure it could be seen that with the reduction of pesticide concentration, the parameters improved. It could be surmised that when the pesticide content increased, a plurality of pesticides molecules competed for the active sites provided by the silver sol. High concentration of pesticides resulted in a severe reduction in the number of active sites. And the other mixture molecules had fewer active sites and their characteristic peak signals were weaker correspondingly. When the pesticide concentration was relatively low, the number of silver sol active sites occupied by pesticide molecules was small. No obvious competitive relationship existed between pesticides. Therefore, one pesticide characteristic peak signal was not interfered by other pesticides. And all the parameters were improved obviously.

It could be seen from Figure 6 that with the concentration of one pesticide decreasing, the correction coefficient of the other pesticide also decreased significantly. It could be deduced that the mutual influence of the functional groups of the two pesticide molecules which weakened each other caused this phenomenon. Concentration of pesticides functional groups reflected the characteristic peaks intensity, in turn reflecting the pesticides concentration. With the increase of one pesticide concentration, the influence of the other pesticide reduced gradually. This phenomenon may exist in other pesticide mixture and correction method would be useful in detecting them accurately.

## 4. Conclusions

Interaction between pesticide mixture components made the Raman spectral detection result lower than the results of single pesticide and affected the accuracy severely. In the study, correction regression models toward different characteristic peaks were built to explore the change rule between pesticide content and characteristic peak intensity of deltamethrin and acetamiprid mixture. The corrected result of single characteristic peak intensity could be used for quantitative calculation to achieve correction of quantitative detection results. And good validation results reflected the accuracy and reliability of the correction. The study demonstrated that great potential existed in this correction method and it could be used in other mixed pesticide detection and extended to mixture detection by Raman spectroscopy. The study provided a novel and feasible correction method in Raman spectral detection.

**Author Contributions:** Conceptualization, Y.P. and Y.L.; methodology, Y.P.; software, K.C. and J.Q.; validation, Y.P. and Y.L.; formal analysis, Y.L.; investigation, Y.L.; resources, Y.P.; data curation, Y.L.; writing—original draft preparation, Y.L.; writing—review and editing, Y.P. and Y.L.; visualization, K.C., J.Q. and Y.L.; supervision, Y.P.; project administration, Y.P.; funding acquisition, Y.P.

**Funding:** This research received no external funding.

**Acknowledgments:** The authors gratefully acknowledge National Key Research and Development Program (2016YFD0400905-05) for providing funding support for this research.

**Conflicts of Interest:** The authors declare no conflict of interest.

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
