# Peer review of "A Correction Method of Mixed Pesticide Content Prediction in Apple by Using Raman Spectra"

_applsci, doi:10.3390/app9081699_

Round 1

Reviewer 1 Report

see attached.

Author Response

Dear Reviewer:

  Thank you for your letter and for your comments concerning our manuscript entitled “A Correction Method of Mixed Pesticide Content Prediction in Apple by Using Raman Spectra”. These comments are all valuable and very helpful for revising and improving our paper, as well as the important guiding significance to our researches. We have studied comments carefully and have made correction which we hope meet with approval. The main corrections in the paper and the responds to the reviewer’s comments are as following:

  Comments:

  1. It is not clear from the text how you calculated the different equations used for the validation of the method and the determination of the pesticides, page 8. Could you explain clearer this method? Could you explain also the correction ratio?

Response 1: We are very sorry for our unclear report in page 8. We have rewritten this part. During the correction, the concentration of the mixed pesticide is estimated by a single pesticide quantitative analysis model to get the estimated pesticide concentration at first. And then the estimated pesticide concentration is applied in correction coefficient model (the equations) to acquire the correction coefficient. The characteristic peak intensity of the pesticide is multiply by the correction coefficient to get the corrected peak intensity which is as high as the peak intensity of single pesticide. Finally the corrected pesticide concentration is obtained by applying corrected peak intensity to quantitative analysis model of the pesticide. We have corrected the sentence and highlighted it with yellow background in Line 252.

2. Could you provide more explanation for Figure 4?

Response 2: It’s our mistake for not clear about Figure 4. We have added more explanation in Line 278 to describe Figure 4 and highlighted it with yellow background.

3. Could you make a table to show the results of the measurements of the apples, with the new method, to show clearly the difference in the two pesticide concentrations?

Response 3: Thank you very much for your suggestion. The table (Table 1) is applied in the new manuscript to show the difference in the two pesticide concentrations.

4. Please replace all figures with better high-quality/resolution images. They are barely distinguishable now.

Response 4: We are sorry for not clear figures in the manuscript. We have replaced all figures. Thank you for your careful reading of our manuscript.

5. Please explain the use of: acetone, silver sol and nitric acid for your method.

Response 5: Your opinion is very helpful to my manuscript. Through our experiments, we found that the surface enhancement effect of applying the silver sol to the epidermis of the sample is not obvious. It is speculated that the silver sol is an aqueous solution, and the solubility of acetamiprid in water is low. The pesticide molecules are adsorbed in the peel. By adding silver sol directly, the acetamiprid molecule is not sufficiently mixed with the silver nanoparticles. Therefore, it’s necessary to dispense an acetone solution before dropping the silver sol to dissolve and disperse the acetamiprid molecule in the acetone solution. Finally, adding nitric acid solution is to increase the number of hydrogen ions (H+) in the vicinity of the silver sol. As the pesticide particles are of negative charge, they are more likely to adsorb near the silver sol. When the pesticide molecule adsorbs or is very close to the nanostructure, its Raman signal is significantly enhanced. Nitric acid shortens the distance between the pesticide molecules and the silver nanoparticles. So the Raman signal is further enhanced. This part has been added in Line 113 in the manuscript with yellow background.

6. Could you also provide additional concentration units, such as ppb or ppm, along with g/kg for comparison?

Response 5: Thank you for your advice. We have add mg/kg(ppb) as concentration unit in new manuscript.

  The new manuscript has been uploaded as a Word file. Special thanks to you for your comments.

  2019-04-13

Reviewer 2 Report

The reviewer appreciates the research effort.  The research approach seems reasonable and the conclusions agree with the methodology used. The main problem is the use of the language.  The paper needs a technical editor fluent in English.  I found myself correcting almost every line in the introduction.  The issue goes throughout the paper making the overall message difficult to understand.  Examples are below:

Line 34-36  Most pesticide potential impacts to human health are either due to processing during agricultural production or the food consumption with high residuals, ingestion, and so on.

Line 36-37 The interaction between different pesticide components has a serious impact on the detection of mixed pesticide residue during raman analysis.  test results.

Lines 37-38 The phenomenon makes the detection result inaccurate and its necessary to find a method to eliminate the interference.

Lines 38-39 Deltamethrin (C22H19Br2NO3) is an efficient synthetic pyrethroid insecticides created in 1974…

Lines 41-43 While deltamethrin is also one of limited used pesticides; several studies had demonstrated that deltamethrin increased the risk of neurodegenerative disorders like Parkinson, Alzheimer disease, developmental deficits and learning disabilities [5-8].

Lines 60-69- Rephrase- Raman spectroscopy is used in almost every sentence.

Lines 106-107 40 apples were chosen randomly as calibration set and the others for validation.

On the content of the paper- did the authors considered comparing the results using gold nanoparticles versus the silver nanoparticles used.   The approach seems reasonable. The following paper may be helpful on how results are presented 

Molecules. 2018 Jun 15;23(6). pii: E1458. doi: 10.3390/molecules23061458. Density Functional Theory Analysis of Deltamethrin and Its Determination in Strawberry by Surface Enhanced Raman Spectroscopy.

Dong T1,2, Lin L3,4, He Y5,6, Nie P7,8,9, Qu F10,11, Xiao S12,13.

Author Response

Dear Reviewer:

Thank you for your letter and for the reviewers’ comments concerning our manuscript entitled “A Correction Method of Mixed Pesticide Content Prediction in Apple by Using Raman Spectra”.

Comments:

The reviewer appreciates the research effort.  The research approach seems reasonable and the conclusions agree with the methodology used. The main problem is the use of the language.  The paper needs a technical editor fluent in English.  I found myself correcting almost every line in the introduction.  The issue goes throughout the paper making the overall message difficult to understand.  Examples are below:

Line 34-36  Most pesticide potential impacts to human health are either due to processing during agricultural production or the food consumption with high residuals, ingestion, and so on.

Line 36-37 The interaction between different pesticide components has a serious impact on the detection of mixed pesticide residue during raman analysis.  test results.

Lines 37-38 The phenomenon makes the detection result inaccurate and it’s necessary to find a method to eliminate the interference.

Lines 38-39 Deltamethrin (C22H19Br2NO3) is an efficient synthetic pyrethroid insecticides created in 1974…

Lines 41-43 While deltamethrin is also one of limited used pesticides; several studies had demonstrated that deltamethrin increased the risk of neurodegenerative disorders like Parkinson, Alzheimer disease, developmental deficits and learning disabilities [5-8].

Lines 60-69- Rephrase- Raman spectroscopy is used in almost every sentence.

Lines 106-107 40 apples were chosen randomly as calibration set and the others for validation.

On the content of the paper- did the authors considered comparing the results using gold nanoparticles versus the silver nanoparticles used.   The approach seems reasonable. The following paper may be helpful on how results are presented

Molecules. 2018 Jun 15; 23(6). pii: E1458. doi: 10.3390/molecules23061458. Density Functional Theory Analysis of Deltamethrin and Its Determination in Strawberry by Surface Enhanced Raman Spectroscopy.

Dong T1,2, Lin L3,4, He Y5,6, Nie P7,8,9, Qu F10,11, Xiao S12,13.

Response:

We are very sorry for the language errors in this manuscript and inconvenience they caused in your reading. The manuscript has been thoroughly revised and edited and highlighted it with yellow background, so we hope it can meet the journal’s standard. Thanks so much for your useful comments.

The Raman spectroscopy enhancement of deltamethrin signal is very well, but the enhancement of acetamiprid is not obvious. Acetamiprid is a kind of organophosphorus pesticide. In our previous, we applied both gold sol and silver sol in organophosphorus pesticide residue with Raman spectral technology. We prepared two groups of chlorpyrifos solution with same concentration. The chlorpyrifos solution was enhanced with gold sol and silver sol respectively. The spectral signal was collected and analyzed and the results were shown in the figure below. The characteristic peak (at 631 cm-1 and 680 cm-1) intensity of chlorpyrifos enhanced by silver nanoparticles was stronger than the ones of gold nanoparticles, which meant the enhancement effect was more obvious. And the fluorescent background of spectrum enhanced by silver nanoparticles was low. And there is some enhancement of silver sol for deltamethrin signal. So silver sol is chosen as enhancer in mixed pesticide residue detection in the study.

The comparison chart of silver sol and gold sol enhancement effect.

That article Density functional theory analysis of deltamethrin and its determination in strawberry by surface enhanced Raman spectroscopy is very helpful to my research and has been cited in new manuscript and added to the Reference.

The new manuscript has been uploaded as a Word file. Special thanks to you for your careful work.

2019.04.13
